# Opposing Actions of Octopamine and Tyramine on Honeybee Vision

**DOI:** 10.3390/biom11091374

**Published:** 2021-09-17

**Authors:** Felix Schilcher, Markus Thamm, Martin Strube-Bloss, Ricarda Scheiner

**Affiliations:** 1Biocenter, Department of Behavioural Physiology and Sociobiology, Julius-Maximilians-Universität of Würzburg, Am Hubland, 97074 Würzburg, Germany; markus.thamm@uni-wuerzburg.de (M.T.); ricarda.scheiner@uni-wuerzburg.de (R.S.); 2Department of Biological Cybernetics and Theoretical Biology, University of Bielefeld, 25, 33615 Bielefeld, Germany; martin.strube-bloss@uni-bielefeld.de

**Keywords:** biogenic amines, neurotransmitter, phototaxis, *Apis mellifera*, ERG, behaviour, modulation, visual system, octopamine, tyramine

## Abstract

The biogenic amines octopamine and tyramine are important neurotransmitters in insects and other protostomes. They play a pivotal role in the sensory responses, learning and memory and social organisation of honeybees. Generally, octopamine and tyramine are believed to fulfil similar roles as their deuterostome counterparts epinephrine and norepinephrine. In some cases opposing functions of both amines have been observed. In this study, we examined the functions of tyramine and octopamine in honeybee responses to light. As a first step, electroretinography was used to analyse the effect of both amines on sensory sensitivity at the photoreceptor level. Here, the maximum receptor response was increased by octopamine and decreased by tyramine. As a second step, phototaxis experiments were performed to quantify the behavioural responses to light following treatment with either amine. Octopamine increased the walking speed towards different light sources while tyramine decreased it. This was independent of locomotor activity. Our results indicate that tyramine and octopamine act as functional opposites in processing responses to light.

## 1. Introduction

The biogenic amines octopamine (OA) and tyramine (TA) play decisive roles in the modulation of honeybee behaviour. They can act as neurohormones, neurotransmitters and neuromodulators [1,2,3,4,5,6,7]. Octopamine is present in the honeybee brain in a concentration of up to 900 pg/brain [8,9]. Tyramine is present in the honeybee brain in much smaller quantities, ranging between 50 and 65 pg/brain [10]. Five different OA receptor genes and two TA receptor genes have been characterized in the honeybee. The activation of the respective receptor proteins can turn on very different intracellular second messenger cascades, which differentially affect physiology and behaviour. *AmOctαR1* leads to a change in intracellular calcium ([Ca^2+^]_i_) when activated [11]. The remaining receptors are coupled with the adenylyl cyclase and lead to changes in intracellular cAMP ([cAMP]_i_) when activated. AmOctαR2 and *AmTAR1* decrease [cAMP]_i_ [12,13], while AmOctβR1/2/3-4 and AmTAR2 increase [cAMP]_i_ [14,15].

Based on the structural similarities of the receptor subtypes and intracellular signalling it is generally assumed that the insect OA/TA system is comparable to the deuterostome epinephrine/norepinephrine system [4,5,7,16,17]. Furthermore, some studies show that OA and TA can have opposite effects, indicating that both amines may act functionally antagonistically [18,19]. Octopamine has frequently been shown to increase responsiveness to different stimulus modalities in honeybees and other insects [1,4,5,6,7,20]. *Drosophila melanogaster* mutants lacking a functional tyramine β hydroxylase, which mediates the final step in OA synthesis, showed a significantly reduced responsiveness to sucrose due to the lack of OA or an increase in TA titres [21]. Additionally, OA plays further important roles in associative learning and memory [1,4,22] by mediating and modulating the reward in appetitive learning [7,23,24]. In contrast to OA, the role of TA has been studied less intensively. The *Drosophila* mutant *honoka* displays a reduced expression of the TAR1 receptor [25]. These mutant flies are slightly hyperactive, have defects in olfactory perception and display a reduced TA-induced muscle contraction. Furthermore, TA was shown to rescue cocaine sensitization defects in *Drosophila* iav/TRPV channel mutants [26].

In the honeybee, OA and TA have been shown to have functionally similar effects in different taste-related behaviours. Both amines accelerate the rate of habituation of the proboscis extension response (PER) [27] and increase responsiveness to sucrose in honeybees [7,28,29]. They can both improve appetitive learning performance in the PER assay, most likely by increasing responsiveness to the reward [10,30]. Further, both OA and TA are involved in the social organisation of honeybees. Octopamine speeds up the adult behavioural maturation [31] and is associated with the transition from hive work to foraging [32]. Additionally, expression of the OA receptor gene *AmOctαR1* is higher in foragers compared to nurse bees [33] and the levels of both biogenic amines are higher in foragers compared to nurse bees as well [9,10]. The TA receptor gene *AmTAR1* is located on a quantitative trait locus linked to foraging behaviour [34]. However, how OA and TA affect division of labour in honeybees is unclear. They assumedly modulate sensory responses to sensory stimuli, i.e., gustatory, visual and olfactory cues, as has been shown for the gustatory system.

While some behavioural evidence suggests that OA plays in honeybee vision, little is known about the role of TA in the visual system. Octopamine was shown to enhance the direction-specific antennal responses during presentation of moving stripe patterns [35]. Whether TA affects this behaviour is unknown. Interestingly, in honeybee nectar foragers we showed earlier that TA enhances phototaxis without affecting locomotor behaviour. Octopamine had the opposite effect, i.e., reducing the walking speed towards a light source [20].

To differentiate effects of both amines on the periphery and on the decision-making processes in the central nervous system, we performed comparative studies using electroretinography (ERG) and phototaxis. To investigate whether the substances have a similar effect on honeybees of different ages or behavioural groups, we used young nurse-aged bees and forager bees.

## 2. Materials and Methods

### 2.1. Honeybees

Honeybees (*Apis mellifera carnica*) were reared at the departmental apiary and collected individually in uncoated bottles with snap-on caps. After collection, the honeybees were placed into cages and maintained in an incubator (30 °C—constant darkness) overnight with access to 50% sugar solution *ad libitum*. Bees were tested for behaviour and/or sensory responses on the following day. Different honeybee groups were chosen for the two experiments. For the ERG experiments, returning non-pollen foragers were caught directly at the hive entrance. For the phototaxis experiments, two groups of honeybees were collected. Foragers were sampled during winter and were therefore collected inside a glasshouse at a feeder containing a 50% sucrose solution. Young honeybees were sampled in spring. Here, we marked newly emerged honeybees and inserted them into an existing colony. They were collected 6 to 14 days later to be used for the experiments, thus representing hive bees. Due to the experimental design and a timespan of 6 months during the experiments, different bee groups had to be used for the different experiments. However, during the same experiment, the two treatment groups were identical.

### 2.2. Electroretinography

For the ERG, a honeybee was removed from the cage, immobilized on ice and mounted on a small acrylic glass block. After mounting, the honeybee’s mandibles and neck were fixed with low melting dental wax. A tiny window was cut into the head capsule between the antenna base and the ocelli. The window could be folded backwards to apply the substances. The glands and trachea dorsal to the honeybee´s brain were removed. A reference electrode (silver wire, diameter: 25 µm; Nilaco, Tokyo, Japan) was placed into one eye and connected to a common ground, whilst the recording electrode, located inside a glass capillary (1B100F-3, WPI, Sarasota, FL, USA) was pulled with a DMZ-Universal Puller as done before [36] and filled with 0.1 M KCl; it was then inserted into the contralateral eye. The signal was amplified 50x (Neuroprobe Amplifier 1600, A-M Systems, Inc, Sequim, WA, USA) and high-pass filtered by an acquisition board (Labtrax 4/16, WPI, Sarasota, FL, USA) and recorded using LabScribe 2 (iWorx Systems Inc., Dover, NH, USA).

A xenon light source (Perkin Elmer optoelectronics, XL2000 Fiber Optic Illumination System) emitting a daylight spectrum between 350 nm and 800 nm was used in combination with three greyscale intensity filters (36%–, 59%– and 100% light intensity) to stimulate the compound eye with a maximum light intensity of 4.95 × 10^16^ photons/cm^2^. To open the light beam we used a manual shutter with a shutter time of 85 ± 6 ms. After placing a honeybee into the ERG setup, it was allowed to rest in constant darkness for 15 min before starting the stimulation with the three intensities in ascending order. Each intensity was applied four times with an inter-trial interval and an inter-stimulus interval of 1 min (Pre-test). Next, the head window was opened and 1 µL of Ringer (270 mM NaCl, 3.2 mM KCl, 1.2 mM CaCl_2_, 10 mM MgCl_2_, 10 mM 3-(*N*-morpholino) propanesulfonic acid, pH 7.4) or 1 µL of either OA or TA in one of three concentrations (10^−3^ mol/L, 10^−4^ mol/L or 10^−5^ mol/L—diluted in Ringer) was applied directly onto the honeybee brain. After another adaptation period of 15 min, the intensities were applied in the same way as in the Pre-test (Post-test). All concentrations for one substance as well as one Ringer control were tested each day using a new honeybee for each substance. The testing order of the substances was distributed randomly. Individual raw ERG data are shown for OA and TA respectively (Appendix A).

### 2.3. Phototaxis Assay

For the phototaxis experiment, a honeybee was removed from the cage, immobilized on ice and mounted in a small plastic tube. Thereafter, the median ocellus was punctured using a small micro dissecting needle. Afterwards, it was injected with 300 nL of Ringer (see above), tyramine (10^−2^ mol/L—diluted in Ringer) or octopamine (10^−2^ mol/L—diluted in Ringer) using a micro manipulator and glass capillaries (1B100F-4, WPI, Sarasota, FL, USA), pulled as described above. Each honeybee was allowed to rest in a Petri dish (diameter = 85 mm) for 15 min in constant darkness. Then it was placed inside the phototaxis arena to measure light responsiveness and walking parameters. The phototaxis assay was analysed as described before [20,36,37]. In short, dark-adapted honeybees were individually placed inside the phototaxis arena and their mean velocity in darkness was recorded for 2 min to measure their locomotor activity. Afterwards, the arena was illuminated with different green light-emitting diodes (LEDs, wavelength = 527 nm) of different light intensities (3%, 6%, 12%, 25%, 50% and 100% light intensity) with a maximum light intensity of 2.61 × 10^14^ photons/cm^2^. For technical reasons, we used non-modifiable LEDs in the phototaxis assay and not the same white light source as in the ERG setup. Two LEDs with the same light intensity were placed opposite each other. Once a honeybee reached a light source, the LED was switched off, and the opposite LED with the same light intensity was switched on. This was repeated four times for each light intensity. Each experiment was started by switching on the LED with the lowest light intensity. The other LEDs were turned on in ascending order. The honeybee´s walking time towards each light source was recorded using a computer stopwatch (Comfort Software Group) [37]. Honeybees for one treatment and the corresponding Ringer controls were tested each day in a pseudo random order from 9 am until 4 pm. Honeybees were kept in the cage with ad libitum access to 50% sucrose solution under constant darkness until injection with the treatment or Ringer.

### 2.4. Data Analysis

For the ERG experiment, data were recorded using LabScribe 2 (iWorx Systems, Inc.). We extracted the maximal response amplitude of the ERG response to compare them before (pre) and after (post) treatment using a one-way ANOVA. Post hoc analyses were conducted using Dunnett’s multiple comparison test. Differences in the ERG responses elicited due to the three light intensity filters were analysed with a repeated measures (RM) one-way ANOVA followed by a Bonferroni´s multiple comparison test. For the phototaxis experiment, the mean velocity during the dark-runs was analysed using a Student’s *t*-test, since data were distributed normally. The means of one honeybee of four trials for each light intensity were calculated, compared and analysed with a RM 2-way ANOVA using GraphPad PRISM (GraphPad Software Inc., V7, San Diego, CA, USA). Post hoc analyses were conducted using Bonferroni´s multiple comparison test.

## 3. Results

### 3.1. The Effect of Octopamine and Tyramine on the ERG Response

Prior to treatment with either amine or the control solution, we investigated whether the three light intensities tested elicited differential receptor responses. In general, the more transparent the filter, the higher the receptor responses independent of the treatment (RM one way ANOVA, *p* < 0.001; for details see Table 1).

Afterwards, we measured the ERG responses of honeybees following treatment with OA, TA or Ringer. Octopamine had a significant overall effect on the ERG response compared to the Ringer control at all three light intensities (Figure 1A–C; Table 2). It significantly increased the ERG response at all three light intensities for a concentration of 10^−3^ mol/L (Table 2). Tyramine had a significant overall effect on the pre-post response compared to the control at two out of three light intensities (Figure 1E–F; Table 2). It significantly decreased the ERG response at 59% light intensity and 100% light intensity for a concentration of 10^−4^ mol/L (Table 2).

### 3.2. Tyramine and Octopamine Have Opposite Effects on the Phototaxis of In-Hive Bees and Foragers

Neither OA (Figure 2A, Table 3) nor TA (Figure 2B, Table 3) affected locomotor behaviour of in-hive bees in the dark arena. Light intensity significantly influenced phototaxis of in-hive bees in both experiments (Figure 2C,D; Table 3). Octopamine significantly decreased the time in-hive bees took to walk towards the switched-on light source compared to the control group (Figure 2C, Table 3), while TA significantly increased the time in-hive bees needed to reach the different switched-on light sources (Figure 2D, Table 3).

Similar to hive bees, no effect of OA (Figure 3A; Table 3) nor TA (Figure 3B; Table 3) could be observed on the mean velocity of foragers in the dark arena. As expected, the factor light intensity significantly influenced phototactic behaviour of foragers in both experiments (Figure 3C,D; Table 3). Octopamine significantly decreased the walking time of foragers towards the switched-on light sources compared to the control group (Figure 3C, Table 3), while TA significantly increased the time foragers needed to reach the different switched-on light sources (Figure 3D, Table 3). 

## 4. Discussion

In this study we investigated the influence of octopamine and tyramine on honeybee responses to light. One major goal was to separate the effects of both amines at the sensory input level (ERG) and the behavioural output level (phototaxis). To understand whether the stronger attraction to light induced by OA was based on a higher perception of light at the sensory periphery we quantified ERG responses. In the ERG, OA (10^−3^ mol/L) elicited stronger receptor responses compared to controls. Tyramine (10^−4^ mol/L) had the opposite effect. The same pattern could be observed on the behavioural level. While OA elicited faster walking behaviour to light, indicating a stronger incentive value of the light, TA had the opposite effect. Neither amine affected the velocity during the dark runs and presumably also during the phototaxis assay. Our findings are in line with earlier studies showing that OA mainly has arousing functions in insects [5,7,32] and that it increases the perceived value of a food source of honeybee foragers [38]. Whether TA has similar or opposing effects is little-known so far. Only a few behavioural experiments were conducted including both biogenic amines. It has been shown that OA can decrease the walking speed towards different light sources when applied chronically, while TA can increase the walking speed in foragers [20]. However, TA also influenced the general locomotor activity in those experiments, so it is impossible to state whether the change in walking times due to TA treatment was a locomotor effect, an effect of increased light perception or a mixture of both in those experiments. The present study differs mainly in the drug application. While we injected both monoamines locally, the authors in the earlier study [20] fed them over three consecutive days before conducting the experiments, possibly resulting in opposite effects. Other studies could also show an arousing effect of OA on the phototaxis. Feeding formamidines, toxins reported to inhibit the OA-stimulated adenylate cyclase, to *Drosophila* reduced phototactic behaviour, indicating a positive effect of OA on the phototaxis [39]. Different application methods could possibly lead to different target receptors. Octopamine and tyramine are known to elicit different effects depending on their targeted receptors [7]. However, due to the lack of suitable antibodies, the location of most OA/TA receptors in the adult honeybee is still unknown. Only the spatial distribution of the octopamine receptor *AmOctαR1* and of the tyramine receptor *AmTAR1* have been described in the honeybee brain. While strong labelling of *AmOctαR1* can be observed for the optic lobes [40], *AmTAR1* is not present here [41]. This indicates that *AmOctαR1* is a strong candidate for the observed effects in this study. In *Drosophila*, Kholy et al. [42] showed expression of *Oamb* and the *TyrRIII* in the optic lobes of *Drosophila melanogaster*. Similar to the *AmOctαR1*, the *Oamb* increases [cAMP]_i_ [43]. However, a *TyrRIII* honeybee homologue, which decreases [cAMP]_i_ [44] is currently unknown. Similar to flies, different OA/TA receptors should be present in the optic lobes of honeybees which might explain our results. Yet, our current experiments do not allow us to specify which receptors are activated by OA or TA. Here we suggest targeted knockout of individual receptor genes using CRISPR/Cas9 [45] or RNA interference [46] in future studies.

Both neurotransmitters are known to modulate not only the central nervous system (CNS) but also peripheral organs expressing respective receptors [47]. It has been shown that OA can target receptors in the CNS, as well as in the periphery, independently of the application method [30,48,49]. Therefore, application duration might be more important than the application itself. Application over three consecutive days might target the CNS and the periphery, while a local injection might preferably target receptors in the honeybee brain. Furthermore, TA is the metabolic precursor of octopamine [4]. Over time, TA might be converted into OA by the enzyme tyramine β hydroxylase. Thus, feeding TA over three consecutive days could lead to an OA effect rather than the expected TA effect. Additionally, a constant treatment of honeybees with either amine might lead to an internalization of the respective receptors which in turn could also lead to the opposite effect [49]. Here, it would be important in future investigations to quantify OA and TA brain titres directly after the phototaxis experiment. In addition to applying the CRISPR/Cas9 technique for a targeted knockout, RNA interference (RNAi) might be an interesting option to reduce receptor gene expression [50,51]. One could argue that an ocellus injection might have a strong negative impact on bee’s behaviour, which might lead to different results. However, we did show that the ocellus injection, does not negatively affect honeybee behaviour during the assay (Appendix A).

To find out whether the opposing effects of OA and TA on walking speed towards light were related to perception, we performed ERG experiments. Octopamine increased the photoreceptor response, while TA decreased it. These results show the same pattern as the phototaxis experiments, leading to the conclusion that sensory input and behavioural output might be directly linked. Little is known about the influence of biogenic amines on the ERG response of honeybees. Lim and Wasserman [52] showed that washing OA-containing seawater over the eye of *Limulus polyphemus* increases the receptor potential in ERG experiments, while Battelle et al. [53] showed that OA increased the ERG amplitude of a *Limulus polyphemus* eye. Erber et al. [54] demonstrated that OA could increase the visual antennal response. However, Chyb et al. [55] found OA to be decreasing the ERG response in *Drosophila melanogaster*. As seen in the previously mentioned behavioural studies, results of biogenic amine experiments can be contradicting. Another explanation might be that OA- and TA receptors are very similar. Tyramine does not only bind to TA receptors but also to OA receptors when applied in high concentrations, although TA has a much higher affinity towards TA receptors [14,15]. If TA bound to all TA receptors present, it might also have activated OA receptors. OA receptors could then elicit different or even opposite effects.

It seems likely that the results obtained in our study reflect the short-term modulation of the visual system in honeybees by OA and TA. Studies that obtained different results also differed in their application methods, indicating a difference between short- and long-term modulation of perception by OA and TA. This is supported by a study from Scheiner et al. [29]. They showed that injecting TA into the abdomen of honeybees leads to an increase in OA and TA in the honeybee brain. This effect is most likely a result of the tyramine β hydroxylase converting TA into OA. While the exact time point of the OA and TA titre quantification is not stated, PER experiments were conducted prior to the amine quantification of the same honeybees. This indicates that at least 1.5 hours passed between the injection and the quantification of both biogenic amines. This time seems sufficient for metabolizing TA into OA.

As stated before, the phototaxis and the ERG results both show the same opposing pattern for octopamine and tyramine. This coincides with the proposed hypothesis by Roeder et al. [4,5,6] stating that the OA/TA system in insects can be compared to the epinephrine/norepinephrine system in deuterostomes. This is also supported by other studies. Saraswati et al. [18] showed the opposing functions of OA and TA in the locomotion of *Drosophila melanogaster* larvae. Furthermore, Fussnecker et al. [56] showed that honeybees spent a significantly increased time flying when treated with OA, whereas those treated with TA spent a significantly decreased time flying compared to controls. Yet, when feeding or injecting OA and TA into the thorax, both increased the PER of honeybees [28]. This indicates that both substances fulfil complementary roles in some sensory systems but opposing functions in other systems. This study clearly shows the opposing functions of OA and TA on the visual system of honeybees. However, further experiments need to be conducted into differentiating long-term and short-term modulation as well as turning off or blocking single receptors to fully understand the modulating effects of both important biogenic amines.

## Figures and Tables

**Figure 1 biomolecules-11-01374-f001:**
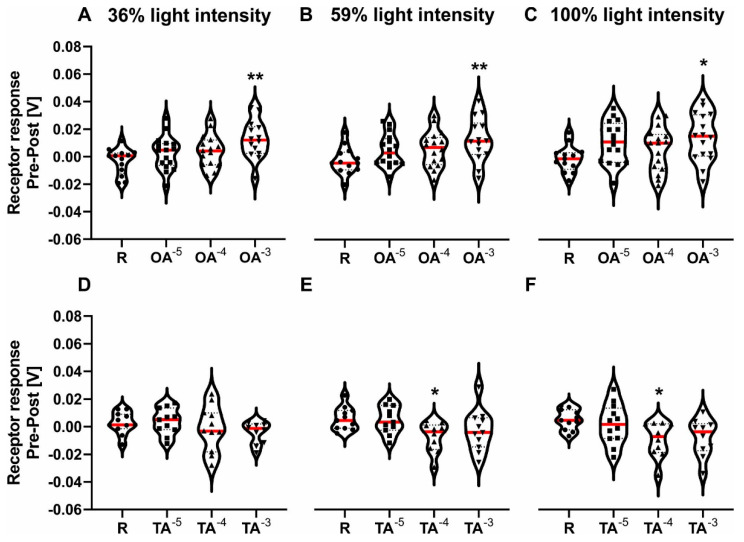
ERG−amplitude receptor response pre-post for three concentrations of octopamine (x-axis in **A**–**C**) and three concentrations of tyramine (x-axis in **D**–**F**) and the respective Ringer control (R). The different light intensities are shown at the top. The median is marked in red. Dots, squares, upper triangles and lower triangles represent individual data points for Ringer, OA/TA 10^−5^ mol/L, OA/TA 10^−4^ mol/L, OA/TA 10^−3^ mol/L, respectively. (**A**–**C**): Pre-post response after OA and Ringer were applied using 36%, 59% and 100% light intensity filters for A, B, and C respectively. A significant overall effect was found for all three light intensities. OA 10^−3^ mol/L significantly increased the amplitude (for statistics, see Table 2). (**D**–**F**): Pre-post responses after TA or Ringer were applied using 36%, 59% and 100% light intensity filters for D, E, and F respectively. No significant differences were found for the 36% light intensity filter. A significant overall effect was found for 59% and 100% light intensity. TA 10^−4^ mol/L significantly decreased the amplitude (for statistics, see Table 2). Significant differences between Ringer and either treatment are indicated by asterisks (* *p* < 0.05, ** *p* < 0.01).

**Figure 2 biomolecules-11-01374-f002:**
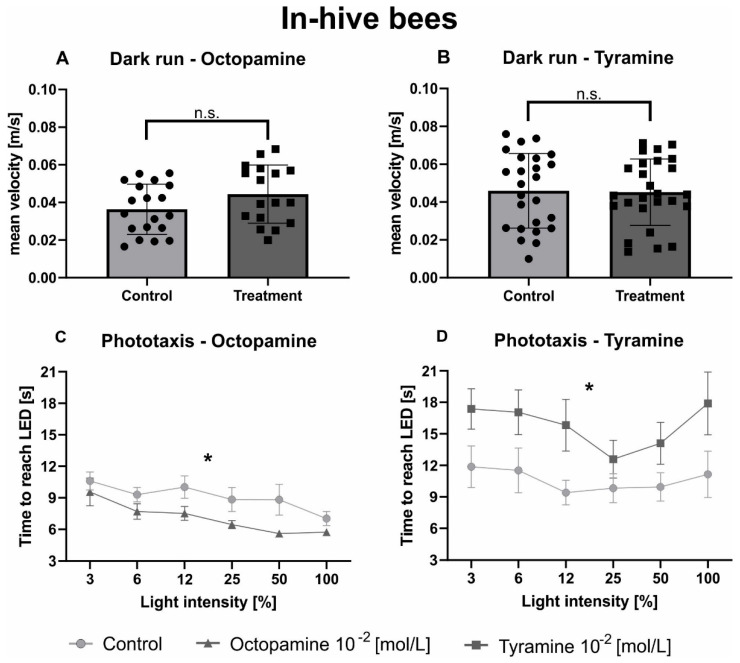
Octopamine and tyramine have opposite effects on the phototaxis of in-hive bees. Control bees are shown in light grey. Those injected with OA (10^−2^ mol/L) or TA (10^−2^ mol/L) are shown in dark grey. (**A**,**B**): Average velocity (mean + standard deviation) of honeybees during one minute of constant movement in the dark arena. Neither OA nor TA differed significantly from the Ringer control in their mean velocity in the dark (Table 3). (**C**,**D**): Average walking time (mean + standard error) towards the different switched-on light sources. The factor light intensity significantly influenced phototactic behaviour (Table 3). Octopamine decreased the time honeybees needed to reach the switched-on LEDs significantly compared to the control solution, while TA increased it (Table 3). Significant differences between Ringer and OA/TA are indicated by asterisks (n.s. *p* > 0.05, * *p* < 0.05).

**Figure 3 biomolecules-11-01374-f003:**
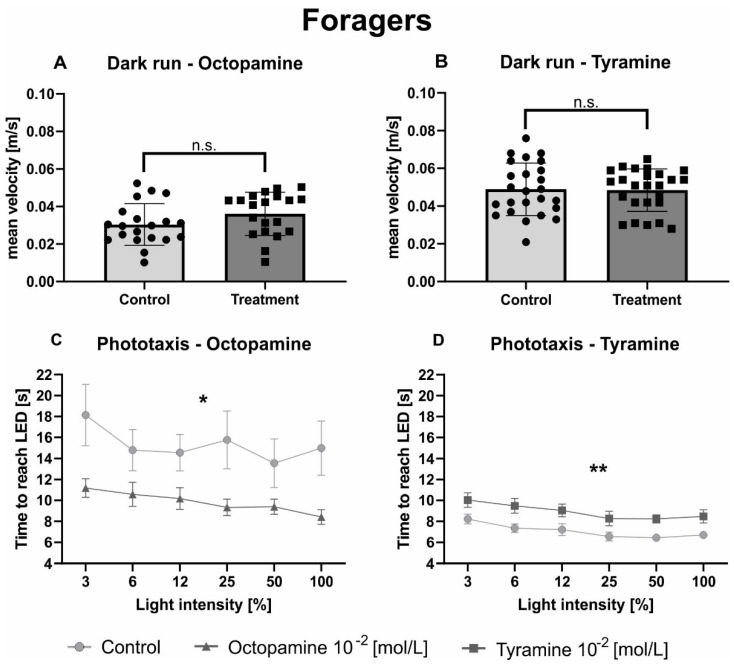
Octopamine and tyramine have opposite effects on the phototaxis of foragers. Control bees are shown in light grey. Those injected with OA (10^−2^ mol/L) or TA (10^−2^ mol/L) are shown in dark grey. (**A**,**B**): Average velocity (mean + standard deviation) of honeybees during one minute of constant movement in the dark arena. Neither OA nor TA did differ significantly from Ringer in their mean velocity in the dark (Table 3). (**C**,**D**): Average walking time (mean + standard error) towards the different switched-on light sources. The light intensity factor significantly influenced phototacic behavior (Table 3). Octopamine decreased the time honeybees needed to reach the switched-on LEDs significantly compared to the control solution, while TA increased it (Table 3). Significant differences between Ringer and OA/TA are indicated by asterisks (n.s. *p* > 0.05, * *p* < 0.05, ** *p* < 0.01).

**Table 1 biomolecules-11-01374-t001:** Analysis of the ERG response of three light intensity filters before and after the application of octopamine/tyramine (treatment) or Ringer (control).

OA—Pre-Response	Friedmann Test		χ^2^_(2)_	*p* Value	
Ringer—Control			14.63	<0.001	***
	Dunn´s test	*n* (Ringer)	*n* (Treatment)		
	36% vs. 59%	16	16	0.024	*
	36% vs. 100%	16	16	<0.001	***
	59% vs. 100%	16	16	0.867	n.s.
OA—post response	Friedmann test		χ^2^_(2)_	*p* value	
Ringer—Control			16.63	<0.001	***
	Dunn´s test	*n* (Ringer)	*n* (Treatment)		
	36% vs. 59%	16	16	0.008	**
	36% vs. 100%	16	16	<0.001	***
	59% vs. 100%	16	16	0.99	n.s.
OA—pre-response	Friedmann test		χ^2^_(2)_	*p* value	
OA—Treatment			26.79	<0.001	***
	Dunn´s test	*n* (Ringer)	*n* (Treatment)		
	36% vs. 59%	48	48	0.001	**
	36% vs. 100%	48	48	<0.001	***
	59% vs. 100%	48	48	0.99	n.s.
OA—post response	Friedmann test		χ^2^_(2)_	*p* value	
OA—Treatment			38.17	<0.001	***
	Dunn´s test	*n* (Ringer)	*n* (Treatment)		
	36% vs. 59%	48	48	<0.001	***
	36% vs. 100%	48	48	<0.001	***
	59% vs. 100%	48	48	0.459	n.s.
TA—pre-response	Friedmann test		χ^2^_(2)_	*p* value	
Ringer—Control			24	<0.001	***
	Dunn´s test	*n* (Ringer)	*n* (Treatment)		
	36% vs. 59%	12	12	0.043	*
	36% vs. 100%	12	12	<0.001	***
	59% vs. 100%	12	12	0.043	*
TA—post response	Friedmann test		χ^2^_(2)_	*p* value	
Ringer—Control			24	<0.001	***
	Dunn´s test	*n* (Ringer)	*n* (Treatment)		
	36% vs. 59%	12	12	0.043	*
	36% vs. 100%	12	12	<0.001	***
	59% vs. 100%	12	12	0.043	*
TA—pre-response	Friedmann test		χ^2^_(2)_	*p* value	
TA—Treatment			70.06	<0.001	***
	Dunn´s test	*n* (Ringer)	*n* (Treatment)		
	36% vs. 59%	36	36	<0.001	***
	36% vs. 100%	36	36	<0.001	***
	59% vs. 100%	36	36	<0.001	***
TA—post response	Friedmann test		χ^2^_(2)_	*p* value	
TA—Treatment			62.39	<0.001	***
	Dunn´s test	*n* (Ringer)	*n* (Treatment)		
	36% vs. 59%	36	36	<0.001	***
	36% vs. 100%	36	36	<0.001	***
	59% vs. 100%	36	36	<0.001	***

Significant differences are indicated by asterisks (n.s. *p* > 0.05, * *p* < 0.05, ** *p* < 0.01, *** *p* < 0.001).

**Table 2 biomolecules-11-01374-t002:** Statistical comparison of the octopamine/tyramine (treatment) or Ringer (control) pre-post receptor response in the honeybee retina.

OA—Pre-Post Response	1way ANOVA			F (3, 54)		*p* Value	
- 36% light intensity				4091		0.011	*
	Dunnett´s test	*n* (Ringer)	*n* (Treatment)	q	DF		
	R vs. OA 10-3	13	14	3457	54	0.003	**
	R vs. OA 10-4	13	15	1667	54	0.237	n.s.
	R vs. OA 10-5	13	16	1.37	54	0.386	n.s.
OA—pre-post response	1way ANOVA			F (3, 55)		*p* value	
- 59% light intensity				3176		0.031	*
	Dunnett´s test	*n* (Ringer)	*n* (Treatment)	q	DF		
	R vs. OA 10-3	13	14	3078	55	0.009	**
	R vs. OA 10-4	13	16	1768	55	0.197	n.s.
	R vs. OA 10-5	13	16	1544	55	0.293	n.s.
OA—pre-post response	1way ANOVA			F (3, 57)		*p* value	
- 100% light intensity				2821		0.0469	*
	Dunnett´s test	*n* (Ringer)	*n* (Treatment)	q	DF		
	R vs. OA 10-3	13	16	2744	57	0.022	*
	R vs. OA 10-4	13	16	1232	57	0.471	n.s.
	R vs. OA 10-5	13	16	2118	57	0.098	n.s.
TA—pre-post response	1way ANOVA			F (3, 39)		*p* value	
- 36% light intensity				1.16		0.337	n.s.
	Dunnett´s test	*n* (Ringer)	*n* (Treatment)	statistics	DF		
	R vs. TA 10-3	11	10	1404	39	0.37	n.s.
	R vs. TA 10-4	11	11	0.892	39	0.703	n.s.
	R vs. TA 10-5	11	11	0.232	39	0.991	n.s.
TA—pre-post response	1way ANOVA			F (3, 39)		*p* value	
- 59% light intensity				3304		0.03	*
	Dunnett´s test	*n* (Ringer)	*n* (Treatment)	q	DF		
	R vs. TA 10-3	11	11	2	39	0.211	n.s.
	R vs. TA 10-4	11	10	2677	39	0.029	*
	R vs. TA 10-5	11	11	0.162	39	0.997	n.s.
TA—pre-post response	1way ANOVA			F (3, 39)		*p* value	
- 100% light intensity				3492		0.025	*
	Dunnett´s test	*n* (Ringer)	*n* (Treatment)	q	DF		
	R vs. TA 10-3	11	10	2373	39	0.059	n.s.
	R vs. TA 10-4	11	10	2702	39	0.027	*
	R vs. TA 10-5	11	12	0.6678	39	0.844	n.s.

Significant differences are indicated by asterisks (n.s. *p* > 0.05, * *p* < 0.05, ** *p* < 0.01).

**Table 3 biomolecules-11-01374-t003:** Statistical comparison of the dark runs and the phototaxis of honeybees (either in-hive bees or foragers) treated with octopamine/tyramine (treatment) or Ringer (control).

Octopamine (In-Hive)	Unpaired *t* Test	Statistic	DF	*n* (Ringer)	*n* (Treatment)	*p* Value	
Dark run		*t* = 1.679	34	19	17	0.102	n.s.
Phototaxis	2way ANOVA						
	Intensity	*F*_(5, 170)_ = 6.131				<0.001	***
	Treatment	*F*_(1, 34)_ = 5.750				0.022	*
	Interaction	*F*_(5, 170)_ = 0.625				0.681	n.s.
	Bonferroni test						
	3%	0.824	204	19	17	0.999	n.s.
	6%	1258	204	19	17	0.999	n.s.
	12%	1960	204	19	17	0.308	n.s.
	25%	1874	204	19	17	0.374	n.s.
	50%	2531	204	19	17	0.073	n.s.
	100%	1017	204	19	17	0.999	n.s.
Tyramine (in-hive)	unpaired *t* test	statistic	DF	*n* (Ringer)	*n* (Treatment)	*p* value	
Dark run		*t* = 0.139	49	25	26	0.890	n.s.
Phototaxis	2way ANOVA						
	Intensity	*F*_(5, 245)_ = 2.564				0.028	*
	Treatment	*F*_(1, 49)_ = 4.919				0.031	*
	Interaction	*F*_(5, 245)_ = 0.669				0.647	n.s.
	Bonferroni test						
	3%	1916	294	25	26	0.338	n.s.
	6%	1931	294	25	26	0.327	n.s.
	12%	2236	294	25	26	0.157	n.s.
	25%	0.964	294	25	26	0.999	n.s.
	50%	1452	294	25	26	0.886	n.s.
	100%	2352	294	25	26	0.116	n.s.
Octopamine (forager)	unpaired *t* test	statistic	DF	*n* (Ringer)	*n* (Treatment)	*p* value	
Dark run		*t* = 1.595	38	20	20	0.119	n.s.
Phototaxis	2way ANOVA						
	Intensity	*F*_(5, 190)_ = 4.342				< 0.001	***
	Treatment	*F*_(1, 38)_ = 5.223				0.028	*
	Interaction	*F*_(5, 190)_ = 1.514				0.187	n.s.
	Bonferroni test						
	3%	2695	228	20	20	0.045	*
	6%	1630	228	20	20	0.627	n.s.
	12%	1700	228	20	20	0.543	n.s.
	25%	2492	228	20	20	0.081	n.s.
	50%	1610	228	20	20	0.653	n.s.
	100%	2545	228	20	20	0.070	n.s.
Tyramine (forager)	unpaired *t* test	statistic	DF	*n* (Ringer)	*n* (Treatment)	*p* value	
Dark run		*t* = 0.123	48	25	25	0.903	n.s.
Phototaxis	2way ANOVA						
	Intensity	*F*_(5, 245)_ = 5.986				< 0.001	***
	Treatment	*F*_(1, 49)_ = 11.29				0.002	**
	Interaction	*F*_(5, 245)_ = 0.072				0.996	n.s.
	Bonferroni test						
	3%	2412	294	26	25	0.099	n.s.
	6%	2838	294	26	25	0.029	*
	12%	2435	294	26	25	0.093	n.s.
	25%	2268	294	26	25	0.144	n.s.
	50%	2377	294	26	25	0.109	n.s.
	100%	2334	294	26	25	0.122	n.s.

Significant differences are indicated by asterisks (n.s. *p* > 0.05, * *p* < 0.05, ** *p* < 0.01, *** *p* < 0.001).

## Data Availability

Correspondence and requests for materials should be addressed to F.S. and R.S.

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
