# Peer review of "Opposing Actions of Octopamine and Tyramine on Honeybee Vision"

_biomolecules, 2021, doi:10.3390/biom11091374_

Round 1
Reviewer 1 Report
The manuscript by Schilcher et al describes A very interesting phenomenon that is significant for several reasons. First, it examines potentially divergent effects of the two monoamines octopamine and tyramine on a defined sensory perception, and second, it correlates effects on a sensory perception with effects on ei associated behavior.
The experimental design is very well chosen and the application of the substances is also adequate, a problem that usually reduces the reproducibility in such studies, but which could be well solved by the extensive experience of the group. The statistical support of the project is also very good and supports the main conclusions of the work.
The only point of criticism that should be addressed in the discussion is that the authors should try to understand the mechanistic aspects of the effect on the ERG. Since no robust expression profiles are available in bees, data from Drosophila should be usable for this purpose.
Author Response
Response to Reviewer 1
We thank reviewer 1 for the constructive feedback and useful comments.
- The only point of criticism that should be addressed in the discussion is that the authors should try to understand the mechanistic aspects of the effect on the ERG. Since no robust expression profiles are available in bees, data from Drosophila should be usable for this purpose.
R: Thank you very much for your review. We added a paragraph explaining the mechanistic aspects (page 11, line 262 -274).
Reviewer 2 Report
This manuscript deals with a timely and re-emerging issue in neuroscience - the action of chemical neuromodulators on sensory physiology and perception. The authors use honeybees as a research model to test the hypotheses that two putatively opposing chemical neuromodulators (octopamine and tyramine) exert distinctly differential influences on (1) the physiology of retinal photoreceptors, and (2) on low-level reflexive behavioral reactions to light (positive phototaxis). The authors performed multiunit field potential electroretinography (ERG) recordings on honeybees as well as performed a behavioral experiment testing honeybee walking speed towards varying intensity light. The animals were injected with varying concentrations of octopamine (OA) or tyramine (TA) to determine how the neurohormonal agents affect processing of light stimuli. OA increased the walking speed towards the light sources whereas tyramine decreased it, which was indicated to be independent of spontaneous locomotor activity. They tested two different ages of honeybees as well: young hive workers (tested in the spring) and foragers (winter). The authors conclude that these two amines act as functional opposites, either increasing (OA) or decreasing (TA) sensory responsiveness and sensory evoked behavior.
The manuscript is relatively well written and clear on a first read. There are several points to consider in order to improve the manuscript and make it appropriate for a neurophysiology audience.
- The presentation of data does not conform to modern physiology standards. Data compression via box-and-whiskers plots, without presenting individual data points for each animal tested, is broadly viewed as obsolete in the age of modern printing capabilities. I would strongly suggest including the data points from individual animals behind the box plots.
- The authors must include raw time series data for the ERG results. The reader must be able to scrutinize the quality of the ERG recordings. I would suggest showing a few raw voltage traces overlaid, plus mean and/or z-score normalization across the population.
- The light stimuli tested in the physiology and behavioral experiments are quite different. A white-light spectrum from 350-800 nm at 36, 59 and 100% intensity is not comparable to a single wavelength (527 nm) at different intensities. At the very least, some discussion is warranted.
- For the behavioral experiments, please detail the time of day and satiety state of the animals - were these variables controlled across treatments? The data presented seem pretty consistent, so I am guessing that these issues were considered.
- (Line 142) The authors recorded the walking times toward each light source with a stopwatch, which is prone to error - some clarifying discussion might be warranted - i.e. potential error bounds of the measurements.
- I would urge the authors to reconsider the strong conclusion: “neither amine affected locomotion” since walking velocity is merely one component of locomotion. There is no mention of gait patterning, kinematics, step/swing phases, etc. which is to say that locomotion as it is considered by the broader literature is much more nuanced than just phototaxic velocity. One might merely focus on the conclusion that photactic walking speed is unaffected. And possibly introduce a statement about other potential influences.
- Figure 1, why are the control groups (Ringer’s pre-test) between the OA and TA so different? Are they statistically significant?
- Figure 2, why do 2C and 2D have different y-axes? (This also applies to Figure 3C and 3D). It obfuscates the point that the authors are trying to make.
Minor
- Figure 1, why are the control groups (Ringer’s pre-test) between the OA and TA so different? Are they statistically significant?
- Error in Figure 1, 1F is stated in the figure as 1D.
Author Response
Response to Reviewer 2
We thank reviewer 2 for the constructive feedback and useful comments.
- The presentation of data does not conform to modern physiology standards. Data compression via box-and-whiskers plots, without presenting individual data points for each animal tested, is broadly viewed as obsolete in the age of modern printing capabilities. I would strongly suggest including the data points from individual animals behind the box plots.
R: We thank the reviewer and have included data points from individual animals in the respective figures (Figure 1,2,3).
- The authors must include raw time series data for the ERG results. The reader must be able to scrutinize the quality of the ERG recordings. I would suggest showing a few raw voltage traces overlaid, plus mean and/or z-score normalization across the population.
R: We agree and included a figure of raw data points in the new supplement figure 1. However, we think that adding the mean would decrease clarity of the figure. Therefore, we only included 4 traces per light intensity per concentration of the individuals.
- The light stimuli tested in the physiology and behavioral experiments are quite different. A white-light spectrum from 350-800 nm at 36, 59 and 100% intensity is not comparable to a single wavelength (527 nm) at different intensities. At the very least, some discussion is warranted.
R: We agree with the reviewer and now discuss this issue in more detail (page 4, line 140 - 142).
- For the behavioral experiments, please detail the time of day and satiety state of the animals - were these variables controlled across treatments? The data presented seem pretty consistent, so I am guessing that these issues were considered.
R: Thank you, we included the requested details (page 4, line 148 - 151)
- (Line 148) The authors recorded the walking times toward each light source with a stopwatch, which is prone to error - some clarifying discussion might be warranted - i.e. potential error bounds of the measurements.
R: Thank you for your comment. Since only one person performed the entire phototaxis experiment, the human error in quantifying the walking times was similar across individuals. Also, we used the mean of four trips per light intensity per bee to reduce possible effects of human errors in time taking(as already stated on page 4, line 145).
- I would urge the authors to reconsider the strong conclusion: “neither amine affected locomotion” since walking velocity is merely one component of locomotion. There is no mention of gait patterning, kinematics, step/swing phases, etc. which is to say that locomotion as it is considered by the broader literature is much more nuanced than just phototaxic velocity. One might merely focus on the conclusion that photactic walking speed is unaffected. And possibly introduce a statement about other potential influences.
R: Thank you for pointing this out. We changed our conclusion accordingly (page 11, line 244-245)
- Figure 1, why are the control groups (Ringer’s pre-test) between the OA and TA so different? Are they statistically significant?
R: Thank you for your comment. Individual differences in honeybee behavioral responses can be quite large, depending on factors such as age, behavioral role, nutritional status, prior experience etc. (see George et al., 2020; Scheiner et al., 2001; Scheiner et al., 2005). It is therefore not surprising that control groups differ between experiments and stresses the importance of using separate control groups for each experiment as well as testing an equal number of treated bees and control bees on the same day.
- Figure 2, why do 2C and 2D have different y-axes? (This also applies to Figure 3C and 3D). It obfuscates the point that the authors are trying to make.
R: We thank the author for the contribution and changed the respective y-axes accordingly (Figure 2,3).
George, E. A., Bröger, A. K., Thamm, M., Brockmann, A. and Scheiner, R. (2020). Inter-individual variation in honey bee dance intensity correlates with expression of the foraging gene. Genes, Brain Behav. 19,.
Scheiner, R., Page, R. E. and Erber, J. (2001). The effects of genotype, foraging role, and sucrose responsiveness on the tactile learning performance of honey bees (Apis mellifera L.). Neurobiol. Learn. Mem. 76, 138–150.
Scheiner, R., Kuritz-Kaiser, A., Menzel, R. and Erber, J. (2005). Sensory responsiveness and the effects of equal subjective rewards on tactile learning and memory of honeybees. Learn. Mem. 12, 626–635.
Reviewer 3 Report
The article “Opposing actions of octopamine and tyramine on honeybee vision”, submitted for consideration for publication in Biomolecules is an interesting study exploring the antagonistic nature of octopamine (OA) and tyramine (TA) in the honey bee visual system. As little is known about how these systems, this is an important contribution. I believe this study and report are of high quality, and I recommend publication of the report. Most of the following specific comments are just minor suggestions for clarification:
Line 100: Please replace “fold” with “folded”.
Line 104: Were the capillary tubes pulled on the DMZ-Universal Puller as described on Line 129? The description should probably be moved here.
Line 116: Were the amines prepared in the same Ringer’s as described on Line 126? The description should probably be moved here.
Line 135: Please replace “LEDS” with “LEDs”.
Line 136: These light intensities are percentages of what?
Line 160: Please state “repeated measures” on first use of “RM”.
Line 163: Is it possible to stretch Table 1 to fit across the page? This would allow its text to be a little bigger and easier to read.
Line 163: Please just double check that there were no copy/paste errors in preparing Table 1. I find it surprising that all the statistical data are exactly the same for the TA Ringer pre response and post response. It is entirely possible, of course, but it is surprising.
Line 163: Similarly, please double check the P-value for the 59% vs 100% tests in the left column. The left column is all pre response, meaning that no amines have been injected into the bees yet, correct? It is surprising then that the P-values can range from 0.99 to <0.001 for the 59% vs 100% tests. Again, entirely possible, but surprising.
Line 172: I am confused as to how the values in Figure 1 are calculated. The OA increased ERG amplitude, and TA reduced ERG amplitude. This would mean OA caused higher voltages to be recorded, and TA caused lower voltages to be recorded, correct? If so, should the labels on the vertical axes be Post-minus-Pre instead of Pre-minus-Post? If it is the latter, would all the values not be inverted?
Line 172: Also Figure 1, might it be clearer to list the concentrations in ascending order (R, OA-5, OA-4, OA-3)?
Lines 192 & 210: In Figures 2 & 3, I suggest you replace the labels on the vertical axes of panels C and D with something like “Walking time [s]” or “Time to reach LED [s]” or something, instead of just “t [sec]” (especially since the SI unit is “s” and not “sec”).
Line 199: You might want to reword this sentence to something indicating it is the time the bee took to reach the far LED. (That is correct, right?) When I first read this, I interpreted “decreased walking times” as the bee just quit walking sooner, as if the amine substantially reduced locomotion.
Lines 208–209: I recommend you reserve this interpretation for the Discussion section.
Line 233: Also, OA appears to increase a forager bee’s perceived value of a food source (Barron et al. 2007, PNAS 104(5):1703-7). Does acute TA treatment in any of these examples produce the opposite effect?
Line 298: Please define “PER”. (You do so below in Line 310).
Line 305: The epinephrine/norepinephrine system is not in the visual systems of deuterostomes, is it?
Line 425: It looks like many of the authors got repeated in Reference #43. (Note: I did not look closely at the other references; this one just jumped out at me.)
Thank you for the opportunity to review this paper. I enjoyed reading it and look forward to seeing it in press.
Author Response
Response to Reviewer 3
We thank reviewer 3 for giving useful comments and criticism.
Line 101: Please replace “fold” with “folded”.
- done
Line 105: Were the capillary tubes pulled on the DMZ-Universal Puller as described on Line 129? The description should probably be moved here.
R: Thank you for your comment, we moved the description to the correct position (page 3, line 102-106)
Line 118: Were the amines prepared in the same Ringer’s as described on Line 131? The description should probably be moved here.
R: Thank you for your comment, we moved the description to the correct position (page 3, line 118)
Line 139: Please replace “LEDS” with “LEDs”.
- done
Line 139: These light intensities are percentages of what?
- We thank the reviewer for the comment. We added the desired information on page 3, line 113 and line 141.
Line 157: Please state “repeated measures” on first use of “RM”.
R: done
Line 170: Is it possible to stretch Table 1 to fit across the page? This would allow its text to be a little bigger and easier to read.
R: Thank you for your comment. We are not able to stretch the table across one page due to the format restrictions of the journal. However, we were able to change the table layout so the data is hopefully easier to read.
Line 170: Please just double check that there were no copy/paste errors in preparing Table 1. I find it surprising that all the statistical data are exactly the same for the TA Ringer pre response and post response. It is entirely possible, of course, but it is surprising.
R: Thank you very much for your comment. We found it very surprising as well. However, we double checked the data and the values shown in Table 1 are indeed correct.
Line 163: Similarly, please double check the P-value for the 59% vs 100% tests in the left column. The left column is all pre response, meaning that no amines have been injected into the bees yet, correct? It is surprising then that the P-values can range from 0.99 to <0.001 for the 59% vs 100% tests. Again, entirely possible, but surprising.
R: We thank the reviewer for the comment. As mentioned above, the values are indeed correct.
Line 179: I am confused as to how the values in Figure 1 are calculated. The OA increased ERG amplitude, and TA reduced ERG amplitude. This would mean OA caused higher voltages to be recorded, and TA caused lower voltages to be recorded, correct? If so, should the labels on the vertical axes be Post-minus-Pre instead of Pre-minus-Post? If it is the latter, would all the values not be inverted?
R: Thank you for your comment. For clarification: OA increases the ERG response leading to more negative voltages, while TA leads to less negative voltages and ringer stays roughly the same. We included an example table in the response that should explain why OA pre-post becomes positive and TA pre-post stays negative.
|
pre |
post |
pre - post |
OA |
-0.05 |
-0.07 |
0.02 |
TA |
-0.05 |
-0.03 |
-0.02 |
Ringer |
-0.05 |
-0.05 |
0 |
Line 172: Also Figure 1, might it be clearer to list the concentrations in ascending order (R, OA-5, OA-4, OA-3)?
- done
Lines 192 & 210: In Figures 2 & 3, I suggest you replace the labels on the vertical axes of panels C and D with something like “Walking time [s]” or “Time to reach LED [s]” or something, instead of just “t [sec]” (especially since the SI unit is “s” and not “sec”).
R: Thank you very much for the response, we adjusted the y-axes in the respective figures (Figure 2,3)
Line 199: You might want to reword this sentence to something indicating it is the time the bee took to reach the far LED. (That is correct, right?) When I first read this, I interpreted “decreased walking times” as the bee just quit walking sooner, as if the amine substantially reduced locomotion.
R: We thank the reviewer for the contribution. We adjusted the wording (page 7, line 199-201; page 8, line 209-211, line 217-219; page 9, line 2,228-230; page 13, line 347).
Lines 208–209: I recommend you reserve this interpretation for the Discussion section.
R: Thank you. We removed the sentence (page 8, line 219-221).
Line 233: Also, OA appears to increase a forager bee’s perceived value of a food source (Barron et al. 2007, PNAS 104(5):1703-7). Does acute TA treatment in any of these examples produce the opposite effect?
R: Thank you, we now discuss this point on page 11 line 244-248.
Line 329: Please define “PER”. (You do so below in Line 310).
R: Thank you for your response. We now define PER in the introduction (page 2, line 56-57).
Line 322: The epinephrine/norepinephrine system is not in the visual systems of deuterostomes, is it?
R: We thank the reviewer for the comment. We removed the statement.
Line 450: It looks like many of the authors got repeated in Reference #38. (Note: I did not look closely at the other references; this one just jumped out at me.)
R: Thank you very much for your response. We checked the references and updated the once that were not correct.